# Towards Reverse-Engineering Black-Box Neural Networks

**Seong Joon Oh, Max Augustin, Bernt Schiele, Mario Fritz**
Max-Planck Institute for Informatics, Saarland Informatics Campus, Saarbrücken, Germany
`{joon,maxaug,schiele,mfritz}@mpi-inf.mpg.de`

## Abstract

Many deployed learned models are black boxes: given input, returns output. Internal information about the model, such as the architecture, optimisation procedure, or training data, is not disclosed explicitly as it might contain proprietary information or make the system more vulnerable. This work shows that such attributes of neural networks can be exposed from a sequence of queries. This has multiple implications. On the one hand, our work exposes the vulnerability of black-box neural networks to different types of attacks – we show that the revealed internal information helps generate more effective adversarial examples against the black box model. On the other hand, this technique can be used for better protection of private content from automatic recognition models using adversarial examples. Our paper suggests that it is actually hard to draw a line between white box and black box models. The code is available at goo.gl/MbYfsv.

## 1 Introduction

Black-box models take a sequence of query inputs, and return corresponding outputs, while keeping internal states such as model architecture hidden. They are deployed as black boxes usually on purpose – for protecting intellectual properties or privacy-sensitive training data. Our work aims at inferring information about the internals of black box models – ultimately turning them into white box models. Such a reverse-engineering of a black box model has many implications. On the one hand, it has legal implications to intellectual properties (IP) involving neural networks – internal information about the model can be proprietary and a key IP, and the training data may be privacy sensitive. Disclosing hidden details may also render the model more susceptible to attacks from adversaries. On the other hand, gaining information about a black-box model can be useful in other scenarios. E.g. there has been work on utilising adversarial examples for protecting private regions (e.g. faces) in photographs from automatic recognisers (Oh et al., 2017). In such scenarios, gaining more knowledge on the recognisers will increase the chance of protecting one's privacy. Either way, it is a crucial research topic to investigate the type and amount of information that can be gained from a black-box access to a model. We make a first step towards understanding the connection between white box and black box approaches – which were previously thought of as distinct classes.

We introduce the term "model attributes" to refer to various types of information about a trained neural network model. We group them into three types: (1) architecture (e.g. type of non-linear activation), (2) optimisation process (e.g. SGD or ADAM?), and (3) training data (e.g. which dataset?). We approach the problem as a standard supervised learning task *applied over models*. First, collect a diverse set of white-box models ("meta-training set") that are expected to be similar to the target black box at least to a certain extent. Then, over the collected meta-training set, train another model ("metamodel") that takes a model as input and returns the corresponding model attributes as output. Importantly, since we want to predict attributes at test time for black-box models, the only information available for attribute prediction is the query input-output pairs. As we will see in the experiments, such input-output pairs allow to predict model attributes surprisingly well.

In summary, we contribute: (1) Investigation of the type and amount of internal information about the black-box model that can be extracted from querying; (2) Novel metamodel methods that not only reason over outputs from static query inputs, but also actively optimise query inputs that can extract more information; (3) Study of factors like size of the meta-training set, quantity and quality

of queries, and the dissimilarity between the meta-training models and the test black box (generalisability); (4) Empirical verification that revealed information leads to greater susceptibility of a black-box model to an adversarial example based attack.

## 2 RELATED WORK

There has been a line of work on extracting and exploiting information from black-box learned models. We first describe papers on extracting information (*model extraction* and *membership inference* attacks), and then discuss ones on attacking the network using the extracted information (*adversarial image perturbations (AIP)*).

*Model extraction* attacks either reconstruct the exact model parameters or build an *avatar model* that maximises the likelihood of the query input-output pairs from the target model (Tramer et al., 2016; Papernot et al., 2017). Tramer et al. (2016) have shown the efficacy of equation solving attacks and the avatar method in retrieving internal parameters of non-neural network models. Papernot et al. (2017) have also used the avatar approach with the end goal of generating adversarial examples. While the avatar approach first assumes model hyperparameters like model family (architecture) and training data, we discriminatively train a metamodel to predict those hyperparameters themselves. As such, our approach is complementary to the avatar approach.

*Membership inference* attacks determine if a given data sample has been included in the training data (Ateniese et al., 2015; Shokri et al., 2017). In particular, Ateniese et al. (2015) also trains a decision tree metamodel over a set of classifiers trained on different datasets. This work goes far beyond only inferring the training data by showing that even the model architecture and optimisation process can be inferred.

Using the obtained cues, one can launch more effective, focused attacks on the black box. We use *adversarial image perturbations* (AIPs) as an example of such attack. AIPs are small perturbations over the input such that the network is mislead. Research on this topic has flourished recently after it was shown that the needed amount of perturbation to completely mislead an image classifier is nearly invisible (Szegedy et al., 2014; Goodfellow et al., 2015; Moosavi-Dezfooli et al., 2017).

Most effective AIPs require gradients of the target network. Some papers proposed different ways to attack black boxes. They can be grouped into three approaches. (1) Approximate gradients by *numerical gradients* (Narodytska & Kasiviswanathan, 2017; Chen et al., 2017). The caveat is that thousands and millions of queries are needed to compute a single AIP, depending on the image size. (2) Use the *avatar approach* to train a white box network that is supposedly similar to the target (Papernot et al., 2016b;a; Hayes & Danezis, 2017). We note again that our metamodel is complementary to the avatar approach – the avatar network hyperparemters can be determined by the metamodel. (3) Exploit *transferability* of adversarial examples; it has been shown that AIPs generated against one network can also fool other networks (Moosavi-Dezfooli et al., 2017; Liu et al., 2017). Liu et al. (2017) in particular have shown that generating AIPs against an ensemble of networks make it more transferable. We show in this work that the AIPs transfer better within an architecture family (e.g. ResNet or DenseNet) than across, and that such a property can be exploited by our metamodel for generating more targetted AIPs.

## 3 METAMODELS

We want to find out the type and amount of internal information about a black-box model that can be revealed from a sequence of queries. We approach this by first building metamodels for predicting model attributes, and then evaluating their performance on black-box models. Our

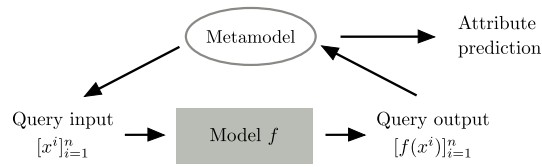

Figure 1: Overview of our approach.

main approach, metamodel, is described in figure 1. In a nutshell, the metamodel is a classifier of classifiers. Specifically, The metamodel submits $n$ query inputs $\left[x^i\right]_{i=1}^n$ to a black box model $f$; the metamodel takes corresponding model outputs $\left[f(x^i)\right]_{i=1}^n$ as an input, and returns predicted model attributes as output. As we will describe in detail, the metamodel not only learns to infer model

attributes from query outputs from a static set of inputs, but also searches for query inputs that are designed to extract greater amount of information from the target models.

In this section, our main methods are introduced in the context of MNIST digit classifiers. While MNIST classifiers are not fully representative of *generic* learned models, they have a computational edge: it takes only five minutes to train each of them with reasonable performance. We could thus prepare a diverse set of 11k MNIST classifiers within 40 GPU days for the meta-training and evaluation of our metamodels. We stress, however, that the proposed approach is generic with respect to the task, data, and the type of models. We also focus on 12 model attributes (table 1) that cover hyperparameters for common neural network MNIST classifiers, but again the range of predictable attributes are not confined to this list.

### 3.1 COLLECTING A DATASET OF CLASSIFIERS

We need a dataset of classifiers to train and evaluate metamodels. We explain how `MNIST-NETS` has been constructed, a dataset of 11k MNIST digit classifiers; the procedure is task and data generic.

#### BASE NETWORK SKELETON

Every model in `MNIST-NETS` shares the same convnet skeleton architecture: "$N$ conv blocks $\rightarrow$ $M$ fc blocks $\rightarrow$ 1 linear classifier". Each conv block has the following structure: "ks $\times$ ks convolution $\rightarrow$ optional $2 \times 2$ max-pooling $\rightarrow$ non-linear activation", where ks (kernel size) and the activation type are to be chosen. Each fc block has the structure: "'linear mapping $\rightarrow$ non-linear activation $\rightarrow$ optional dropout" This convnet structure already covers many LeNet (LeCun et al., 1998) variants, one of the best performing architectures on MNIST[1].

#### INCREASING DIVERSITY

In order to learn generalisable features, the metamodel needs to be trained over a diverse set of models. The base architecture described above already has several free parameters like the number of layers ($N$ and $M$), the existence of dropout or max-pooling layers, or the type of non-linear activation.

Apart from the architectural hyperparameters, we increase diversity along two more axes – optimisation process and the training data. Along the optimisation axis, we vary optimisation algorithm (SGD, ADAM, or

Table 1: MNIST classifier attributes. *Italicised* attributes are derived from other attributes.

| | Code | Attribute | Values |
|---|---|---|---|
| Architecture | act | Activation | ReLU, PReLU, ELU, Tanh |
| | drop | Dropout | Yes, No |
| | pool | Max pooling | Yes, No |
| | ks | Conv ker. size | 3, 5 |
| | #conv | #Conv layers | 2, 3, 4 |
| | #fc | #FC layers | 2, 3, 4 |
| | *#par* | *#Parameters* | $2^{14}, \cdots, 2^{21}$ |
| | ens | Ensemble | Yes, No |
| Opt. | alg | Algorithm | SGD, ADAM, RMSprop |
| | bs | Batch size | 64, 128, 256 |
| Data | split | Data split | $\text{All}_0$, $\text{Half}_{0/1}$, $\text{Quarter}_{0/1/2/3}$ |
| | *size* | *Data size* | All, Half, Quarter |

RMSprop) and the training batch size (64, 128, 256). We also consider training MNIST classifiers on either on the entire MNIST training set ($\text{All}_0$, 60k), one of the two disjoint halves ($\text{Half}_{0/1}$, 30k), or one of the four disjoint quarters ($\text{Quarter}_{0/1/2/3}$, 15k).

See table 1 for the comprehensive list of 12 model attributes altered in `MNIST-NETS`. The number of trainable parameters (#par) and the training data size (size) are not directly controlled but derived from the other attributes. We also augment `MNIST-NETS` with ensembles of classifiers (ens), whose procedure will be described later.

#### SAMPLING AND TRAINING

The number of all possible combinations of controllable options in table 1 is $18, 144$. We also select random seeds that control the initialisation and training data shuffling from $\{0, \cdots, 999\}$, resulting in $18, 144, 000$ unique models. Training such a large number of models is intractable; we have sampled (without replacement) and trained $10, 000$ of them. All the models have been trained with

---

[1] http://yann.lecun.com/exdb/mnist/

learning rate $0.1$ and momentum $0.5$ for 100 epochs. It takes around 5 minutes to train each model on a GPU machine (GeForce GTX TITAN); training of 10k classifiers has taken 40 GPU days.

### PRUNING AND AUGMENTING

In order to make sure that `MNIST-NETS` realistically represents commonly used MNIST classifiers, we have pruned low-performance classifiers (validation accuracy $< 98\%$), resulting in $8,582$ classifiers. Ensembles of trained classifiers have been constructed by grouping the identical classifiers (modulo random seed). Given $t$ identical ones, we have augmented `MNIST-NETS` with $2, \cdots, t$ combinations. The ensemble augmentation has resulted in $11,282$ final models. See appendix table 6 for statistics of attributes – due to large sample size all the attributes are evenly covered.

### TRAIN-EVAL SPLITS

Attribute prediction can get arbitrarily easy by including the black-box model (or similar ones) in the meta-training set. We introduce multiple splits of `MNIST-NETS` with varying requirements on generalization. Unless stated otherwise, every split has $5,000$ training (meta-training), $1,000$ testing (black box), and $5,282$ leftover models.

The Random (R) split randomly (uniform weights) assigns training and test splits, respectively. Under the R split, the training and test models come from the same distribution. We introduce harder Extrapolation (E) splits. We separate a few attributes between the training and test splits. They are designed to simulate more difficult domain gaps when the meta-training models are significantly different from the black box. Specific examples of E splits will be shown in §4.

## 3.2 METAMODEL METHODS

The metamodel predicts the attribute of a black-box model $g$ in the test split by submitting $n$ query inputs and observing the outputs. It is trained over meta-training models $f$ in the training split ($f \sim \mathcal{F}$). We propose three approaches for the metamodels – we collectively name them `kennen`[2]. See figure 2 for an overview.

### KENNEN-O: REASON OVER OUTPUT

`kennen-o` first selects a fixed set of queries $[x^i]_{i=1\cdots n}$ from a dataset. Both during training and testing, always these queries are submitted. `kennen-o` learns a classifier $m_\theta$ to map from the order-sensitively concatenated $n$ query outputs, $[f(x^i)]_{i=1\cdots n}$ ($n \times 10$ dim for MNIST), to the simultaneous prediction of 12 attributes in $f$. The training objective is:

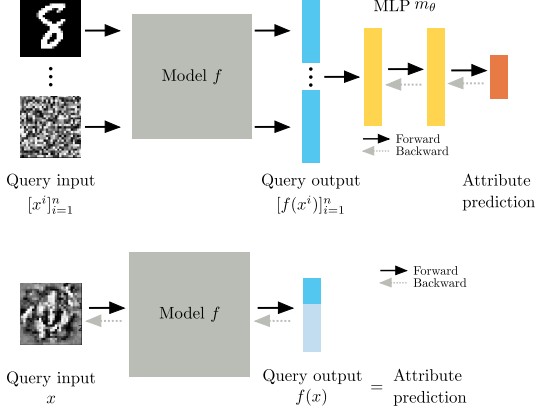

$$\min_{\theta} \ \mathbb{E}_{f \sim \mathcal{F}} \left[ \sum_{a=1}^{12} \mathcal{L} \left( m_\theta^a \left( [f(x^i)]_{i=1}^n \right), y^a \right) \right] \tag{1}$$

where $\mathcal{F}$ is the distribution of meta-training models, $y^a$ is the ground truth label of attribute $a$, and $\mathcal{L}$ is the cross-entropy loss. With the learned parameter $\tilde{\theta}$, $m_{\tilde{\theta}}^a \left( [g(x^i)]_{i=1}^n \right)$ gives the prediction of attribute $a$ for the black box $g$.

Figure 2: Training procedure for metamodels `kennen-o` (top) and `kennen-i` (bottom).

In our experiments, we model the classifier $m_\theta$ via multilayer perceptron (MLP) with two hidden layers with 1000 hidden units. The last layer consists of 12 parallel linear layers for a simultaneous prediction of the attributes. In our preliminary experiments, MLP has performed better than the linear classifiers. The optimisation problem in equation 1 is solved via SGD by approximating the expectation over $f \sim \mathbb{F}$ by an empirical sum over the training split classifiers for 200 epochs.

---

[2]*kennen* means "to know" in German, and "to dig out" in Korean.

For query inputs, we have used a random subset of $n$ images from the validation set (both for MNIST and ImageNet experiments). The performance is not sensitive to the choice of queries (see appendix §C). Next methods (`kennen-i/io`) describe how to actively craft query inputs, potentially outside the natural image distribution.

Note that `kennen-o` can be applied to any type of model (e.g. non-neural networks) with any output structure, as long as the output can be embedded in an Euclidean space. We will show that this method can effectively extract information from $f$ even if the output is a top-k ranking.

### KENNEN-I: CRAFT INPUT

`kennen-i` crafts a *single* query input $\tilde{x}$ over the meta-training models that is trained to repurpose a digit classifier $f$ into a model attribute classifier for a *single* attribute $a$. The crafted input drives the classifier to leak internal information via digit prediction. The learned input is submitted to the test black-box model $g$, and the attribute is predicted by reading off its digit prediction $g(\tilde{x})$. For example, `kennen-i` for max-pooling layer prediction crafts an input $x$ that is predicted as "1" for generic MNIST digit classifiers with max-pooling layers and "0" for ones without. See figure 3 for visual examples.

We describe in detail how `kennen-i` learns this input. The training objective is:

$$\min_{x:\,\text{image}} \mathbb{E}_{f \sim \mathcal{F}} \left[ \mathcal{L}\left(f(x), y^a\right) \right] \qquad (2)$$

where $f(x)$ is the 10-dimensional output of the digit classifier $f$. The condition $x$ : image ensures the input stays a valid image $x \in [0,1]^D$ with image dimension $D$. The loss $\mathcal{L}$, together with the attribute label $y^a$ of $f$, guides the digit prediction $f(x)$ to reveal the attribute $a$ instead. Note that the optimisation problem is identical

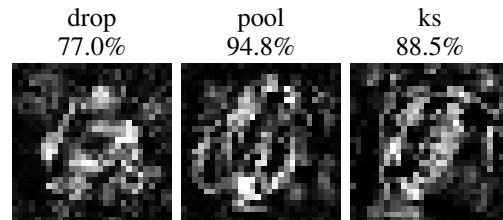

| drop | pool | ks |
|------|------|-----|
| 77.0% | 94.8% | 88.5% |

Figure 3: Inputs designed to extract internal details from MNIST digit classifiers. E.g. feeding the middle image reveals the existence of a max-pooling layer with 94.8% chance.

to the training of digit classifiers except that the ground truth is the attribute label rather than the digit label, that the loss is averaged over the models instead of the images, and that the input $x$ instead of the model $f$ is optimised. With the learned query input $\tilde{x}$, the attribute for the black box $g$ is predicted by $g(\tilde{x})$. In particular, we do not use gradient information from $g$.

We initialise $x$ with a random sample from the MNIST validation set (random noise or uniform gray initialisation gives similar performances), and run SGD for 200 epochs. For each iteration $x$ is truncated back to $[0,1]^D$ to enforce the constraint.

While being simple and effective, `kennen-i` can only predict a single attribute at a time, and cannot predict attributes with more than 10 classes (for digit classifiers). `kennen-io` introduced below overcomes these limitations. `kennen-i` may also be unrealistic when the exploration needs to be stealthy: it submits unnatural images to the system. Also unlike `kennen-o`, `kennen-i` requires end-to-end differentiability of the training models $f \sim \mathcal{F}$, although it still requires only black-box access to test models $g$.

### KENNEN-IO: COMBINED APPROACH

We overcome the drawbacks of `kennen-i` that it can only predict one attribute at a time and that the number of predictable classes by attaching an additional interpretation module on top of the output. Our final method `kennen-io` combines `kennen-i` and `kennen-o` approaches: both input generator and output interpreters are used. Being able to reason over multiple query outputs via MLP layers, `kennen-io` supports the optimisation of multiple query inputs as well.

Specifically, the `kennen-io` training objective is given by:

$$\min_{[x^i]_{i=1}^n:\,\text{images}} \min_{\theta} \mathbb{E}_{f \sim \mathcal{F}} \left[ \sum_{a=1}^{12} \mathcal{L}\left(m_\theta^a\left([f(x^i)]_{i=1}^n\right), y^a\right) \right]. \qquad (3)$$

Note that the formulation is identical to that for `kennen-o` (equation 1), except that the second minimisation problem regarding the query inputs is added. With learned parameters $\tilde{\theta}$ and $[\tilde{x}^i]_{i=1}^n$,

Table 2: Comparison of metamodel methods. See table 1 for the full names of attributes. 100 queries are used for every method below, except for `kennen-i` which uses a single query. The "Output" column shows the output representation: "prob" (vector of probabilities for each digit class), "ranking" (a sorted list of digits according to their likelihood), "top-1" (most likely digit), or "bottom-1" (least likely digit).

| Method | Output | architecture | | | | | | | | optim | | data | | |
|---|---|---|---|---|---|---|---|---|---|---|---|---|---|---|
| | | act | drop | pool | ks | #conv | #fc | #par | ens | alg | bs | size | split | avg |
| Chance | - | 25.0 | 50.0 | 50.0 | 50.0 | 33.3 | 33.3 | 12.5 | 50.0 | 33.3 | 33.3 | 33.3 | 14.3 | 34.9 |
| `kennen-o` | prob | 80.6 | 94.6 | 94.9 | 84.6 | 67.1 | 77.3 | 41.7 | 54.0 | 71.8 | 50.4 | 73.8 | 90.0 | 73.4 |
| `kennen-o` | ranking | 63.7 | 93.8 | 90.8 | 80.0 | 63.0 | 73.7 | 44.1 | **62.4** | 65.3 | 47.0 | 66.2 | 86.6 | 69.7 |
| `kennen-o` | bottom-1 | 48.6 | 80.0 | 73.6 | 64.0 | 48.9 | 63.1 | 28.7 | 52.8 | 53.6 | 41.9 | 45.9 | 51.4 | 54.4 |
| `kennen-o` | top-1 | 31.2 | 56.9 | 58.8 | 49.9 | 38.9 | 33.7 | 19.6 | 50.0 | 36.1 | 35.3 | 33.3 | 30.7 | 39.5 |
| `kennen-i` | top-1 | 43.5 | 77.0 | 94.8 | 88.5 | 54.5 | 41.0 | 32.3 | 46.5 | 45.7 | 37.0 | 42.6 | 29.3 | 52.7 |
| `kennen-io` | score | **88.4** | **95.8** | **99.5** | **97.7** | **80.3** | **80.2** | **45.2** | 60.2 | **79.3** | **54.3** | **84.8** | **95.6** | **80.1** |

the attribute $a$ for the black box $g$ is predicted by $m_\theta^a\left([g(\tilde{x}^i)]_{i=1}^n\right)$. Again, we require end-to-end differentiability of meta-training models $f$, but only the black-box access for the test model $g$.

To improve stability against covariate shift, we initialise $m_\theta$ with `kennen-o` for 200 epochs. Afterwards, gradient updates of $[x^i]_{i=1}^n$ and $\theta$ alternate every 50 epochs, for 200 additional epochs.

# 4 REVERSE-ENGINEERING BLACK-BOX MNIST DIGIT CLASSIFIERS

We have introduced a procedure for constructing a dataset of classifiers (`MNIST-NETS`) as well as novel metamodels (`kennen` variants) that learn to extract information from black-box classifiers. In this section, we evaluate the ability of `kennen` to extract information from black-box MNIST digit classifiers. We measure the *class-balanced* attribute prediction accuracy for each attribute $a$ in the list of 12 attributes in table 1.

ATTRIBUTE PREDICTION

See table 2 for the main results of our metamodels, `kennen-o/i/io`, on the Random split. Unless stated otherwise, metamodels are trained with $5,000$ training split classifiers.

Given $n = 100$ queries with probability output, `kennen-o` already performs far above the random chance in predicting 12 diverse attributes (73.4% versus 34.9% on average); neural network output indeed contains rich information about the black box. In particular, the presence of dropout (94.6%) or max-pooling (94.9%) has been predicted with high precision. As we will see in §4.3, outputs of networks trained with dropout layers form clusters, explaining the good prediction performance.

It is surprising that optimisation details like algorithm (71.8%) and batch size (50.4%) can also be predicted well above the random chance (33.3% for both). We observe that the training data attributes are also predicted with high accuracy (71.8% and 90.0% for size and split).

COMPARING METHODS `KENNEN-O/I/IO`

Table 2 shows the comparison of `kennen-o/i/io`. `kennen-i` has a relatively low performance (average 52.7%), but `kennen-i` relies on a cheap resource: 1 query with single-label output. `kennen-i` is also performant at predicting the kernel size (88.5%) and pooling (94.8%), attributes that are closely linked to spatial structure of the input. We conjecture `kennen-i` is relatively effective for such attributes. `kennen-io` is superior to `kennen-o/i` for all the attributes with average accuracy 80.1%.

## 4.1 FACTOR ANALYSIS

We examine potential factors that contribute to the successful prediction of black box internal attributes. We measure the prediction accuracy of our metamodels as we vary (1) the number of meta-training models, (2) the number of queries, and (3) the quality of query output.

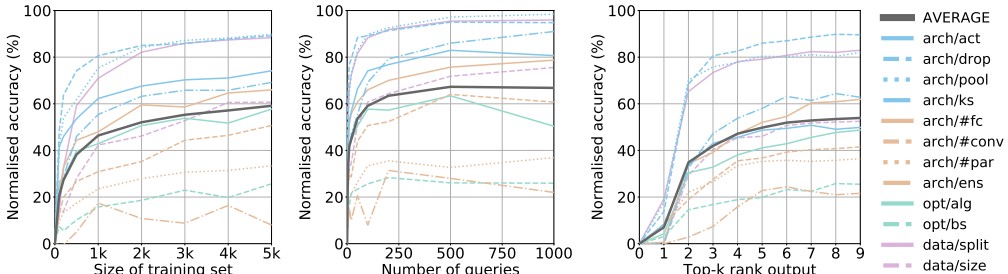

Figure 4: `kennen-o` performance of against the size of meta-training set (left), number of queries (middle), and quality of queries (right). Unless stated otherwise, we use 100 probability outputs and 5k models to train `kennen-o`. Each curve is linearly scaled such that random chance (0 training data, 0 query, or top-0) performs 0%, and the perfect predictor performs 100%.

NUMBER OF TRAINING MODELS

We have trained `kennen-o` with different number of the meta-training classifiers, ranging from 100 to $5,000$. See figure 4 (left) for the trend. We observe a diminishing return, but also that the performance has not saturated – collecting larger meta-training set will improve the performance.

NUMBER OF QUERIES

See figure 4 (middle) for the `kennen-o` performance against the number of queries with probability output. The average performance saturates after $\sim 500$ queries. On the other hand, with only $\sim 100$ queries, we already retrieve ample information about the neural network.

QUALITY OF OUTPUT

Many black-box models return top-k ranking (e.g. Facebook face recogniser), or single-label output. We represent top-k ranking outputs by assigning exponentially decaying probabilities up to $k$ digits and a small probability $\epsilon$ to the remaining.

See table 2 for the `kennen-o` performance comparison among 100 probability, top-10 ranking, bottom-1, and top-1 outputs, with average accuracies 73.4%, 69.7%, 54.4%, and 39.5%, respectively. While performance drops with coarser outputs, when compared to random chance (34.9%), 100 single-label bottom-1 outputs already leak a great amount of information about the black box (54.4%). It is also notable that bottom-1 outputs contain much more information than do the top-1 outputs; note that for high-performance classifiers top-1 predictions are rather uniform across models and thus have much less freedom to leak auxiliary information. Figure 4 (right) shows the interpolation from top-1 to top-10 (i.e. top-9) ranking. We observe from the jump at $k = 2$ that the second likely predictions (top-2) contain far more information than the most likely ones (top-1). For $k \geq 3$, each additional output label exhibits a diminishing return.

## 4.2 WHAT IF THE BLACK-BOX IS QUITE DIFFERENT FROM META-TRAINING MODELS?

So far we have seen results on the Random (R) split. In realistic scenarios, the meta-training model distribution may not be fully covering possible black box models. We show how damaging such a scenario is through Extrapolation (E) split experiments.

EVALUATION

E-splits split the training and testing models based on one or more attributes (§3.1). For example, we may assign shallower models (#layers $\leq 10$) to the training split and deeper ones (#layers ¿10) to the testing split. In this example, we refer to #layers as the *splitting attribute*. Since for an E-split, some classes of the splitting attributes have zero training examples, we only evaluate the prediction accuracies over the non-splitting attributes. When the set of splitting attributes is $\tilde{A}$, a subset of the entire attribute set $A$, we define *E-split accuracy* or E.Acc($\tilde{A}$) to be the mean prediction accuracy over the non-splitting attributes $A \setminus \tilde{A}$. For easier comparison, we report the *normalised accuracy*

(N.Acc) that shows the how much percentage of the R-split accuracy is achieved in the E-split setup on the non-splitting attributes $A \setminus \tilde{A}$. Specifically:

$$\text{N.Acc}(\tilde{A}) = \frac{\text{E.Acc}(\tilde{A}) - \text{Chance}(\tilde{A})}{\text{R.Acc}(\tilde{A}) - \text{Chance}(\tilde{A})} \times 100\% \tag{4}$$

where $\text{R.Acc}(\tilde{A})$ and $\text{Chance}(\tilde{A})$ are the means of the R-split and Chance-level accuracies over $A \setminus \tilde{A}$. Note that N.Acc is 100% if the E-split performance is at the level of R-split and 0% if it is at chance level.

### RESULTS

The normalised accuracies for R-split and multiple E-splits are presented in table 3. We consider three axes of choices of splitting attributes for the E-split: architecture (#conv and #fc), optimisation (alg and bs), and data (size). For example, "E-#conv-#fc" row presents results when metamodel is trained on shallower nets (2 or 3 conv/fc layers each) compared to the test black box model (4 conv and fc layers each).

Not surprisingly, E-split performances are lower than R-split ones (N.Acc $< 100\%$); it is advisable to cover all the expected black-box attributes during meta-training. Nonetheless, E-split performances of `kennen-io` are still far above the chance level (N.Acc $\geq 70\% \gg 0\%$); failing to cover a few attributes during meta-training is not too damaging.

Table 3: Normalised accuracies (see text) of `kennen-o` and `kennen-io` on R and E splits. We denote E-split with splitting attributes *attr1* and *attr2* as "E-*attr1*-*attr2*". Splitting criteria are also shown. When there are two splitting attributes, the first attribute inherits the previous row criteria.

| Split | Train | Test | kennen- o | kennen- io |
|---|---|---|---|---|
| R | - | - | 100 | 100 |
| E-#conv | 2,3 | 4 | 87.5 | 92.0 |
| E-#conv-#fc | 2,3 | 4 | 77.1 | 80.7 |
| E-alg | SGD,ADAM | RMSprop | 83.0 | 88.5 |
| E-alg-bs | 64,128 | 256 | 64.2 | 70.0 |
| E-split | Quarter$_{0/1}$ | Quarter$_{2/3}$ | 83.5 | 89.3 |
| E-size | Quarter | Half,All | 81.7 | 86.8 |
| Chance | - | - | 0.0 | 0.0 |

Comparing `kennen-o` and `kennen-io` for their generalisability, we observe that `kennen-io` consistently outperforms `kennen-o` under severe extrapolation (around 5 pp better N.Acc). It is left as a future work to investigate the intriguing fact that utilising out-of-domain query inputs improves the generalisation of metamodel.

### 4.3 WHY AND HOW DOES METAMODEL WORK?

It is surprising that metamodels can extract inner details with great precision and generalisability. This section provides a glimpse of *why* and *how* this is possible via metamodel input and output analyses. Full answers to those questions is beyond the scope of the paper.

### METAMODEL INPUT (T-SNE)

We analyse the inputs to our metamodels (i.e. query outputs from black-box models) to convince ourselves that the inputs do contain discriminative features for model attributes. As the input is high dimensional (1000 when the number of queries is $n = 100$), we use the t-SNE (van der Maaten & Hinton, Nov 2008) visualisation method. Roughly speaking, t-SNE embeds high dimensional data points onto the 2-dimensional plane such that the pairwise distances are best respected. We then colour-code the embedded data points according to the model attributes. Clusters of same-coloured points indicate highly discriminative features.

The visualisation of input data points are shown in Appendix figures 9 and 10 for `kennen-o` and `kennen-io`, respectively. For experimental details, see Appendix §D. In the case of `kennen-o`, we observe that some attributes form clear clusters in the input space – e.g. Tanh in act, binary dropout attribute, and RMSprop in alg. For the other attributes, however, it seems that the clusters are too complicated to be represented in a 2-dimensional space. For `kennen-io` (figure 10), we observe improved clusters for pool and ks. By submitting crafted query inputs, `kennen-io` induces query outputs to be better clustered, increasing the chance of successful prediction.

METAMODEL OUTPUT (CONFUSION MATRIX)

We show confusion matrices of `kennen-o/io` to analyse the failure modes. See Appendix figures 11 and 12. For `kennen-o` and `kennen-io` alike, we observe that the confusion occurs more frequently with similar classes. For attributes #conv and #fc, more confusion occurs between $(2, 3)$ or $(3, 4)$ than between $(2, 4)$. A similar trend is observed for #par and bs. This is a strong indication that (1) there exists semantic attribute information in the neural network outputs (e.g. number of layers, parameters, or size of training batch) and (2) the metamodels learn semantic information that can generalise, as opposed to merely relying on artifacts. This observation agrees with a conclusion of the extrapolation experiments in §4.2: the metamodels generalise.

Compared to those of `kennen-o`, `kennen-io` confusion matrices exhibit greater concentration of masses both on the correct class (diagonals) and among similar attribute classes (1-off diagonals for #conv, #fc, #par, bs, and size). The former re-confirms the greater accuracy, while the latter indicates the improved ability to extract more semantic and generalisable features from the query outputs. This, again, agrees with §4.2: `kennen-io` generalises better than `kennen-o`.

## 4.4 DISCUSSION

We have verified through our novel `kennen` metamodels that black-box access to a neural network exposes much internal information. We have shown that only 100 single-label outputs already reveals a great deal about a black box. When the black-box classifier is quite different from the meta-training classifiers, the performance of our best metamodel – `kennen-io`– decreases; however, the prediction accuracy for black box internal information is still surprisingly high.

# 5 REVERSE-ENGINEERING AND ATTACKING IMAGENET CLASSIFIERS

While MNIST experiments are computationally cheap and a massive number of controlled experiments is possible, we provide additional ImageNet experiments for practical implications on realistic image classifiers. In this section, we use `kennen-o` introduced in §3 to predict a single attribute of black-box ImageNet classifiers – the architecture family (e.g. ResNet or VGG?). In this section, we go a step further to use the extracted information to attack black boxes with adversarial examples.

## 5.1 DATASET OF IMAGENET CLASSIFIERS

It is computationally prohibitive to train $O(10k)$ ImageNet classifiers from scratch as in the previous section. We have resorted to 19 PyTorch[3] pretrained ImageNet classifiers. The 19 classifiers come from five families: **S**queezenet, **V**GG, VGG-**B**atchNorm, **R**esNet, and **D**enseNet, each with 2, 4, 4, 5, and 4 variants, respectively (Iandola et al., 2016; Simonyan & Zisserman, 2015; Ioffe & Szegedy, 2015; He et al., 2016; Huang et al., 2017). See Appendix table 7 for the the summary of the 19 classifiers. We observe both large intra-family diversity and small inter-family separability in terms of #layers, #parameters, and performances. The family prediction task is not as trivial as e.g. simply inferring the performance.

## 5.2 CLASSIFIER FAMILY PREDICTION

We predict the classifier family (S, V, B, R, D) from the black-box query output, using the method `kennen-o`, with the same MLP architecture (§3). `kennen-i` and `kennen-io` have not been used for computational reasons, but can also be used in principle. We conduct 10 cross validations (random sampling of single test network from each family) for evaluation. We also perform 10 random sampling of the queries from ImageNet validation set. In total 100 random tries are averaged.

Results: compared to the random chance (20.0%), 100 queries result in high `kennen-o` performance (90.4%). With $1,000$ queries, the prediction performance is even 94.8%.

---

[3]`https://github.com/pytorch`

## 5.3 ATTACKING IMAGENET CLASSIFIERS

In this section we attack ImageNet classifiers with adversarial image perturbations (AIPs). We show that the knowledge about the black box architecture family makes the attack more effective.

### ADVERSARIAL IMAGE PERTURBATION (AIP)

AIPs are carefully crafted additive perturbations on the input image for the purpose of misleading the target model to predict wrong labels (Goodfellow et al., 2015). Among variants of AIPs, we use efficient and robust GAMAN (Oh et al., 2017). See appendix figure 7 for examples of AIPs; the perturbation is nearly invisible.

### TRANSFERABILITY OF AIPS

Typical AIP algorithms require gradients from the target network, which is not available for a black box. Mainly three approaches for generating AIPs against black boxes have been proposed: (1) numerical gradient, (2) avatar network, or (3) transferability. We show that our metamodel strengthens the transferability based attack.

We hypothesize and empirically show that AIPs transfer better within the architecture family than across. Using this property, we first predict the family of the black box (e.g. ResNet), and then generate AIPs against a few instances in the family (e.g. ResNet101, ResNet152). The generation of AIPs against multiple targets has been proposed by Liu et al. (2017), but we are the first to systemically show that AIPs generalise better within a family when they are generated against multiple instances from the same family.

Table 4: Transferability of adversarial examples within and across families. We report *misclassification rates*.

| Gen | Target family | | | | |
|---|---|---|---|---|---|
| | S | V | B | R | D |
| Clean | 38 | 32 | 28 | 30 | 29 |
| S | 64 | 49 | 45 | 39 | 35 |
| V | 62 | 96 | 96 | 57 | 52 |
| B | 50 | 85 | 95 | 47 | 44 |
| R | 64 | 72 | 78 | 87 | 77 |
| D | 58 | 63 | 70 | 76 | 90 |
| Ens | 70 | 93 | 93 | 75 | 80 |

We first verify our hypothesis that AIPs transfer better within a family. Within-family: we do a leave-one-out cross validation – generate AIPs using all but one instances of the family and test on the holdout. Not using the exact test black box, this gives a lower bound on the within-family performance. Across-family: still leave out one random instance from the generating family to match the generating set size with the within-family cases. We also include the use-all case (Ens): generate AIPs with one network from *each* family.

See table 4 for the results. We report the *misclassification rate*, defined as $100 -$ top-1 accuracy, on 100 random ImageNet validation images. We observe that the within-family performances dominate the across-family ones (diagonal entries versus the others in each row); if the target black box family is identified, one can generate more effective AIPs. Finally, trying to target all network ("Ens") is not as effective as focusing resources (diagonal entries).

### METAMODEL ENABLES MORE EFFECTIVE ATTACKS

We empirically show that the reverse-engineering enables more effective attacks. We consider multiple scenarios. "White box" means the target model is fully known, and the AIP is generated specifically for this model. "Black box" means the exact target is unknown, but we make a distinction when the family is known ("Family black box").

See table 5 for the misclassification rates (MC) in different scenarios. When the target is fully specified (white box), MC is 100%. When neither the exact target nor the family is known, AIPs are generated against multiple families (82.2%). When the reverse-engineering takes place, and AIPs are generated over the predicted family, attacks become more effective (85.7%). We almost reach the family-oracle case (86.2%).

Table 5: Black-box ImageNet classifier misclassification rates (MC) for different approaches.

| Scenario | Generating nets | MC(%) |
|---|---|---|
| White box | Single white box | 100.0 |
| Family black box | GT family | 86.2 |
| **Black box whitened** | **Predicted family** | **85.7** |
| Black box | Multiple families | 82.2 |

## 5.4 DISCUSSION

Our metamodel can predict architecture families for ImageNet classifiers with high accuracy. We additionally show that this reverse-engineering enables more focused attack on black-boxes.

## 6 CONCLUSION

We have presented first results on the inference of diverse neural network attributes from a sequence of input-output queries. Our novel metamodel methods, kennen, can successfully predict attributes related not only to the architecture but also to training hyperparameters (optimisation algorithm and dataset) even in difficult scenarios (e.g. single-label output, or a distribution gap between the meta-training models and the target black box). We have additionally shown in ImageNet experiments that reverse-engineering a black box makes it more vulnerable to adversarial examples.

### ACKNOWLEDGMENTS

This research was supported by the German Research Foundation (DFG CRC 1223). We thank Seong Ah Choi for her help with the method names, graphics, and colour palettes.

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

APPENDIX

## A    MNIST-NETS STATISTICS

We show the statistics of MNIST-NETS, our dataset of MNIST classifiers, in table 6.

## B    MORE KENNEN-IO RESULTS

We complement the kennen-o results in the main paper (figure 4) with kennen-io results. See figure 5. Similarly for kennen-o, kennen-io shows a diminishing return as the number of training models and the number of queries increase. While the performance saturates with $1,000$ queries, it does not fully saturate with $5,000$ training samples.

## C    ON FINDING THE OPTIMAL SET OF QUERIES

kennen-o selects a random set of queries from MNIST validation set (§3.2). We measure the sensitivity of kennen-o performance with respect to the choice of queries, and discuss the possibility to optimise the set of queries.

With 1, 10, or 100 queries, we have trained kennen-o with 100 independent samples of query sets. The mean and standard deviations are shown in figure 6. The sensitivity is greater for smaller number of queries, but still minute (1.2 pp standard deviation).

Instead of solving the combinatorial problem of finding the optimal set of query inputs from a dataset, we have proposed kennen-io that efficiently solves a continuous optimisation problem to find a set of query inputs from the entire input space. We have compared kennen-io against kennen-o with multiple query samples in figure 6. We observe that kennen-io is better than kennen-o with all 100 query set samples at each level.

We remark that there exists a trade-off between detectability and effectiveness of exploration. While kennen-io extracts information from target model more effectively, it increases the detectability of attack by submitting out-of-domain inputs. If it is possible to optimise or sample the set of natural queries from a dataset or distribution of natural inputs, it will be a strong attack; developing such a method would be an interesting future work.

## D    T-SNE VISUALISATION OF METAMODEL INPUTS

We describe the detailed procedure for the metamodel input visualisation experiment (discussed in §4.3). First, 1000 test-split (Random split) black-box models are collected. For each model, 100 query images are passed (sampled at random from MNIST validation set), resulting in $100 \times 10$ dimensional input data points. We have used t-SNE(van der Maaten & Hinton, Nov 2008) to embed the data points onto the 2-dimensional plane. Each data point is coloured according to each attribute class. The results for kennen-o and kennen-io are shown in figures 9 and 10. Since t-SNE is sensitive to initialisation, we have run the embedding ten times with different random initialisations; the qualitative observations are largely identical.

## E    VISUAL EXAMPLES OF AIPS

In this section, we show examples of AIPs. See figure 7 for the examples of AIPs and the perturbed images. The perturbation is nearly invisible to human eyes. We have also generated AIPs with respect to a diverse set of architecture families (S, V, B, R, D, SVBRD) at multiple $L_2$ norm levels. See figure 8; the same image results in a diverse set of patterns depending on the architecture family.

Table 6: Distribution of attributes in `MNIST-NETS`, and attribute-wise classification performance (on MNIST validation set). Observe that the attributes are evenly distributed and the corresponding classification accuracies also do not correlate much with the attributes. We thus make sure that the classification accuracy alone cannot be a strong cue for predicting attributes.

| | arch/act | | | | arch/drop | | arch/pool | | arch/ks | | arch/#conv | | | arch/#fc | | |
| --- | --- | --- | --- | --- | --- | --- | --- | --- | --- | --- | --- | --- | --- | --- | --- | --- |
| | Tanh | PReLU | ReLU | ELU | Yes | No | Yes | No | 5 | 3 | 2 | 3 | 4 | 2 | 3 | 4 |
| Ratio | 24.8 | 24.9 | 25.3 | 25.1 | 49.8 | 50.3 | 49.9 | 50.2 | 50.3 | 49.7 | 34.0 | 33.4 | 32.7 | 33.1 | 33.5 | 33.4 |
| max | 99.4 | 99.4 | 99.5 | 99.4 | 99.5 | 99.4 | 99.4 | 99.5 | 99.5 | 99.4 | 99.4 | 99.4 | 99.5 | 99.4 | 99.4 | 99.5 |
| median | 98.6 | 98.7 | 98.7 | 98.7 | 98.7 | 98.6 | 98.7 | 98.5 | 98.7 | 98.6 | 98.6 | 98.7 | 98.7 | 98.7 | 98.6 | 98.6 |
| mean | 98.6 | 98.7 | 98.7 | 98.7 | 98.7 | 98.6 | 98.7 | 98.6 | 98.7 | 98.6 | 98.6 | 98.7 | 98.7 | 98.7 | 98.6 | 98.6 |
| min | 98.0 | 98.0 | 98.0 | 98.0 | 98.0 | 98.0 | 98.0 | 98.0 | 98.0 | 98.0 | 98.0 | 98.0 | 98.0 | 98.0 | 98.0 | 98.0 |

| | opt/alg | | | opt/bs | | | data/size | | |
| --- | --- | --- | --- | --- | --- | --- | --- | --- | --- |
| | RMSprop | ADAM | SGD | 64 | 128 | 256 | all | half | quarter |
| Ratio | 33.8 | 32.5 | 33.7 | 32.9 | 33.6 | 33.7 | 14.8 | 28.5 | 56.8 |
| max | 99.2 | 99.4 | 99.5 | 99.3 | 99.4 | 99.5 | 99.5 | 99.3 | 99.1 |
| median | 98.6 | 98.7 | 98.7 | 98.6 | 98.7 | 98.7 | 99.0 | 98.8 | 98.5 |
| mean | 98.6 | 98.7 | 98.7 | 98.6 | 98.7 | 98.6 | 98.9 | 98.8 | 98.5 |
| min | 98.0 | 98.0 | 98.0 | 98.0 | 98.0 | 98.0 | 98.0 | 98.0 | 98.0 |

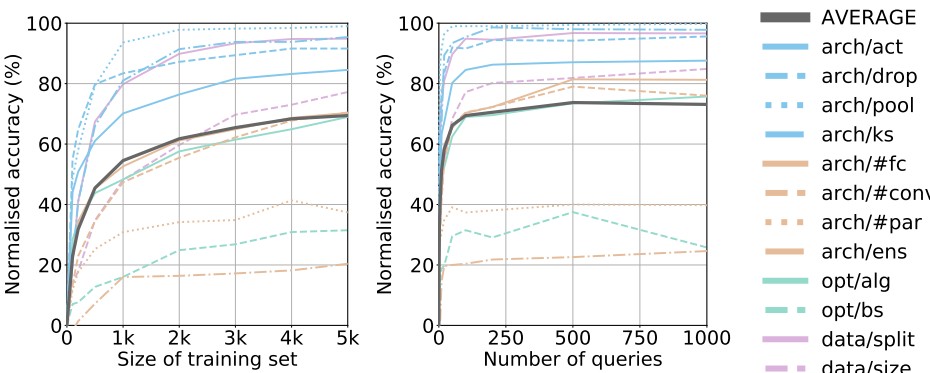

Figure 5: Performance of `kennen-io` with different number of queries (Left) and size of training set (Right). The curves are linearly scaled per attribute such that random chance performs 0%, and perfect predictor performs 100%.

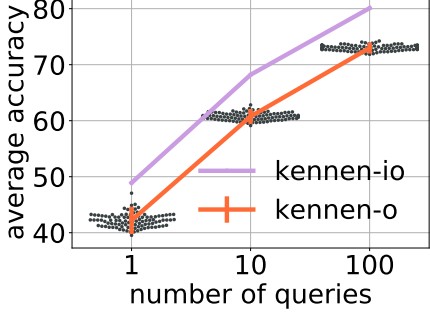

Figure 6: `kennen-o/io` performance at different number of queries. `kennen-o` is shown with 100 independent query samples per level (black dots) – the dots are spread horizontally for visualisation purpose. Their mean (curve) and ±2 standard deviations (error bars) are also shown.

Table 7: Details of ImageNet classifiers. We describe each family **S**queezenet, **V**GG, VGG-**B**atchNorm, **R**esNet, and **D**enseNet verbally, and show key model statistics for each member in the family. We observe intra-family diversity (e.g. R) and inter-family similarity (e.g. between V and B) in terms of the top-5 validation error and the number of trainable parameters.

| | S (2016) | | V (2014) | | | | B (2015) | | | | R (2015) | | | | | D (2016) | | | |
|---|---|---|---|---|---|---|---|---|---|---|---|---|---|---|---|---|---|---|---|
| Description | Lightweight convnet | | Conv layers followed by fc layers | | | | VGG with batch normalisation | | | | Very deep convnet with residual connections | | | | | ResNet with dense residual connections | | | |
| Members | v1.0 | v1.1 | 11 | 13 | 16 | 19 | 11 | 13 | 16 | 19 | 18 | 34 | 50 | 101 | 152 | 121 | 161 | 169 | 201 |
| #layers | 26 | 26 | 11 | 13 | 16 | 19 | 11 | 13 | 16 | 19 | 21 | 37 | 54 | 105 | 156 | 121 | 161 | 169 | 201 |
| $\log_{10}$ #params | 6.1 | 6.1 | 8.1 | 8.1 | 8.1 | 8.2 | 8.1 | 8.1 | 8.1 | 8.2 | 7.1 | 7.3 | 7.4 | 7.6 | 7.8 | 6.9 | 7.3 | 7.5 | 7.2 |
| Top-1 error | 41.9 | 41.8 | 31.0 | 30.1 | 28.4 | 27.6 | 29.6 | 28.5 | 26.6 | 25.8 | 30.2 | 26.7 | 23.9 | 22.6 | 21.7 | 25.4 | 24.0 | 22.8 | 22.4 |
| Top-5 error | 19.6 | 19.4 | 11.4 | 10.8 | 9.6 | 9.1 | 10.2 | 9.6 | 8.5 | 8.2 | 10.9 | 8.6 | 7.1 | 6.4 | 5.9 | 7.8 | 6.2 | 7.0 | 6.4 |

Original  Perturbation  Perturbed  Original  Perturbation  Perturbed

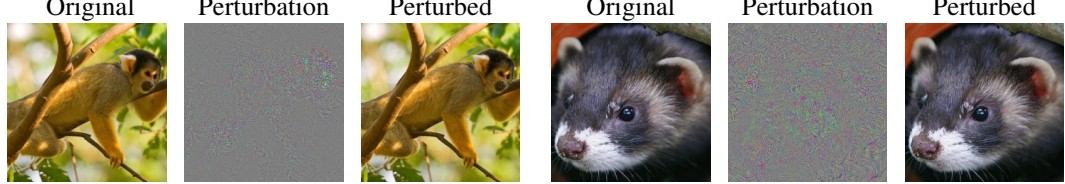

Figure 7: AIP for an ImageNet classifier. The perturbations are generated at $L_2 = 1 \times 10^{-4}$.

Original Image

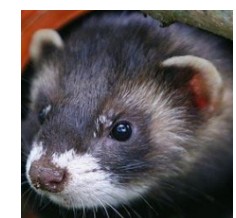

$L_2 = 1 \times 10^{-5}$     $L_2 = 2 \times 10^{-5}$     $L_2 = 5 \times 10^{-5}$     $L_2 = 1 \times 10^{-4}$

**S**queezeNet

VGG

VGG-**B**atchNorm

**R**esNet

DenseNet

All (SVBRD)

Figure 8: Adversarial perturbations for the same input image (top) generated with diverse ImageNet classifier families (S, V, B, R, D, SVBRD) at different norm constraints. The perturbation images are normalised at the maximal perturbation for visualisation. We observe diverse patterns across classifier families within the same $L_2$ ball.

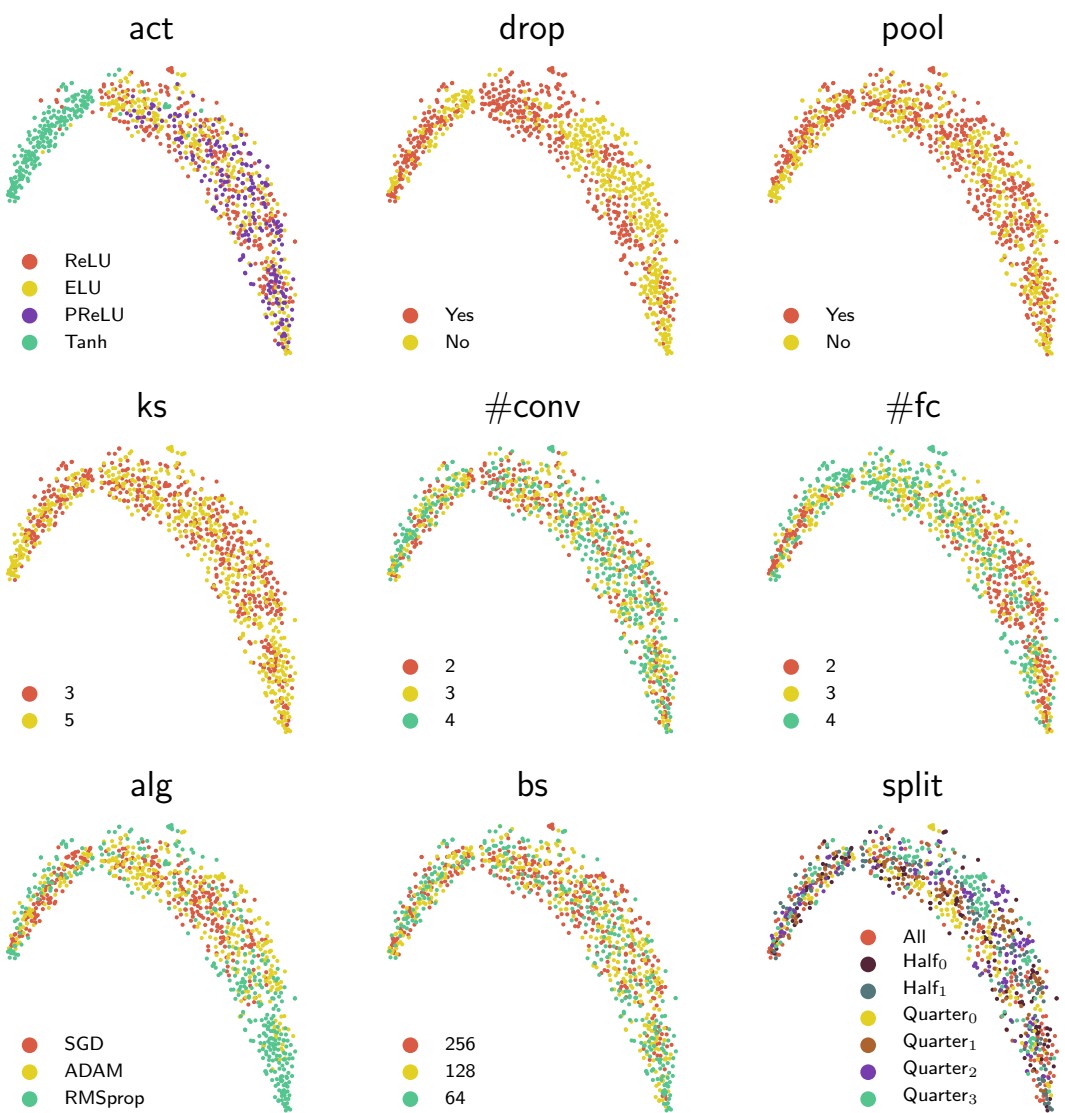

Figure 9: Probability query output embedded into 2-D plane via t-SNE. The same embedding is shown with different colour-coding for each attribute. These are the inputs to the `kennen-o` metamodel.

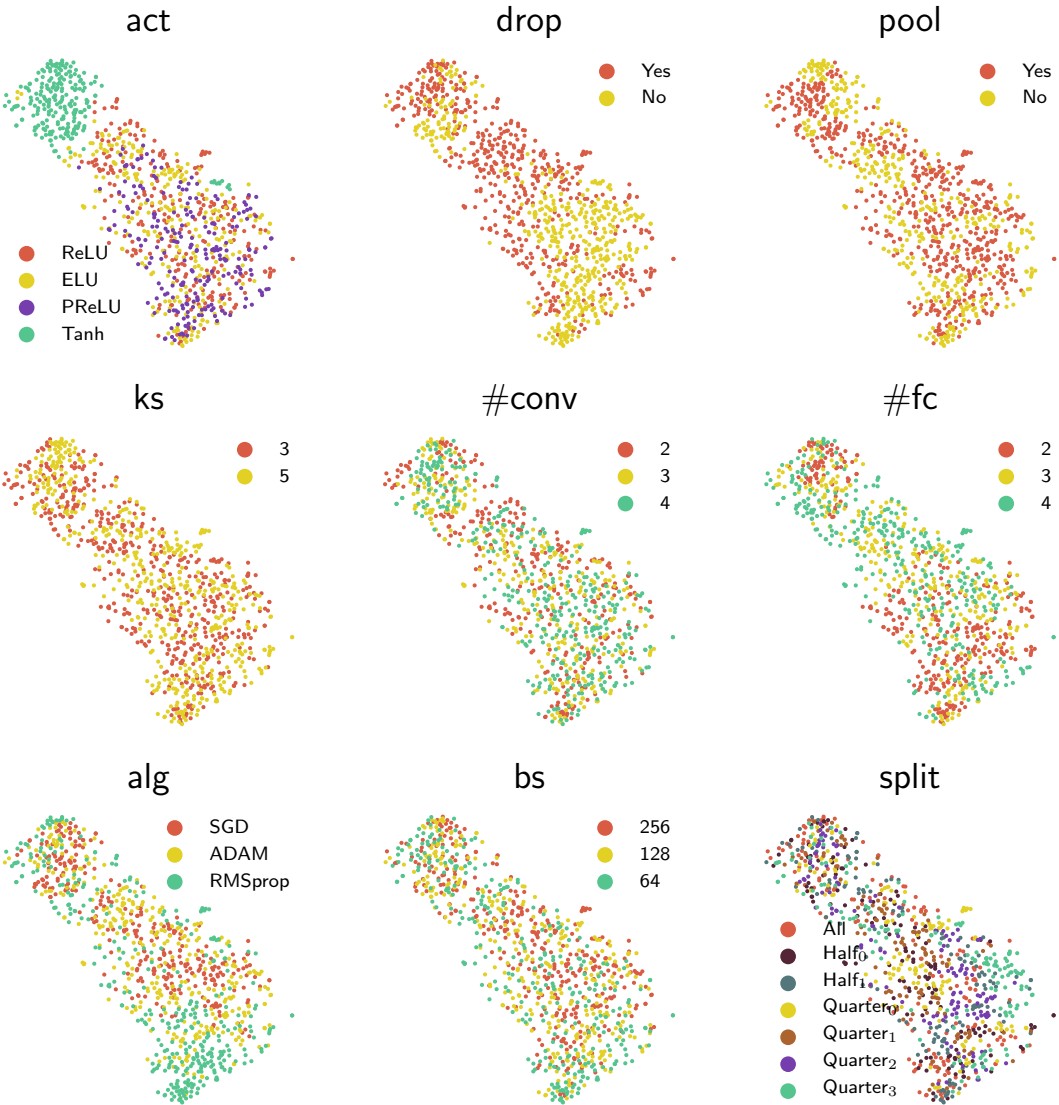

Figure 10: Probability query output embedded into 2-D plane via t-SNE. The same embedding is shown with different colour-coding for each attribute. These are the inputs to the `kennen-io` metamodel.

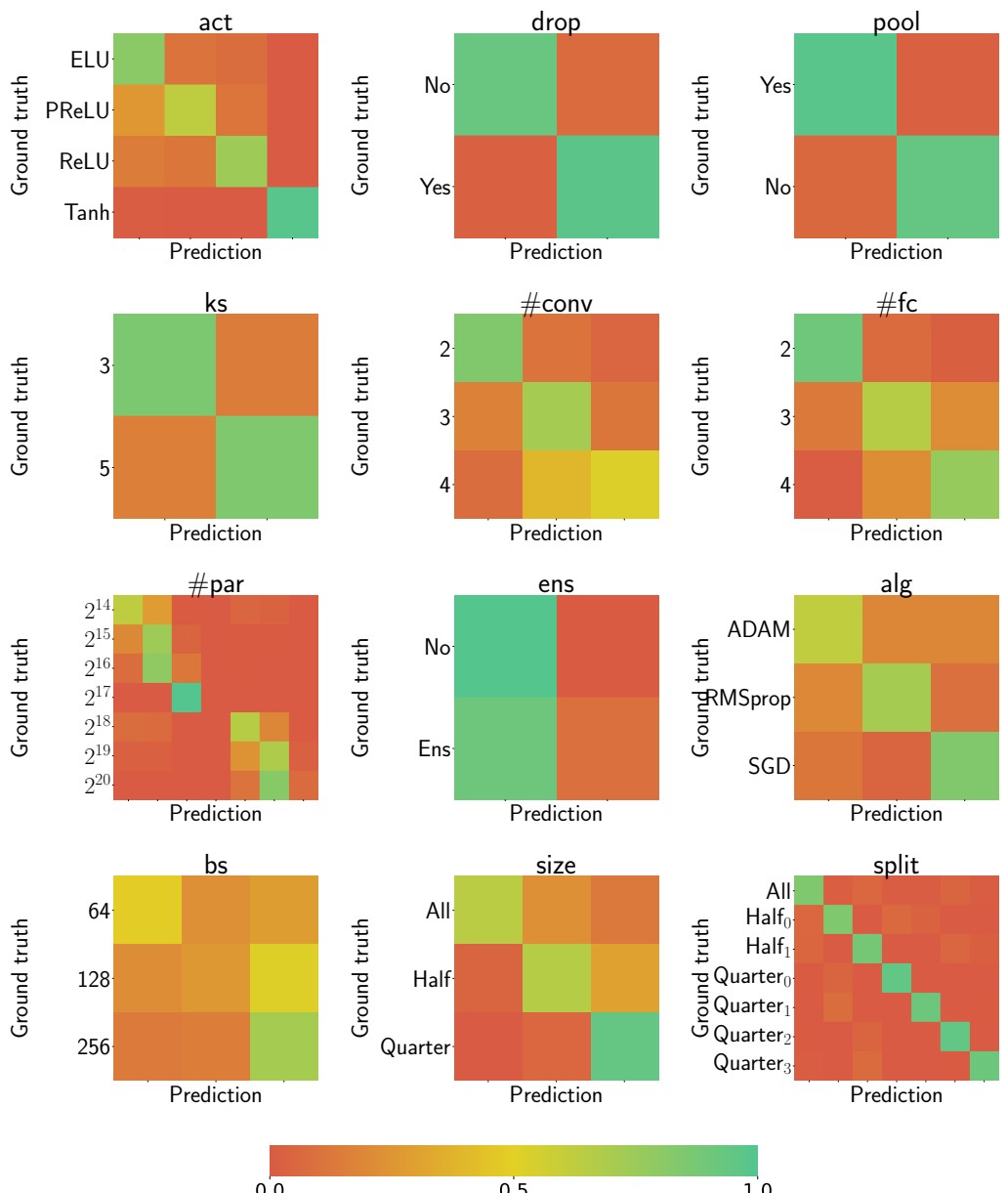

Figure 11: Confusion matrices for `kennen-o`.

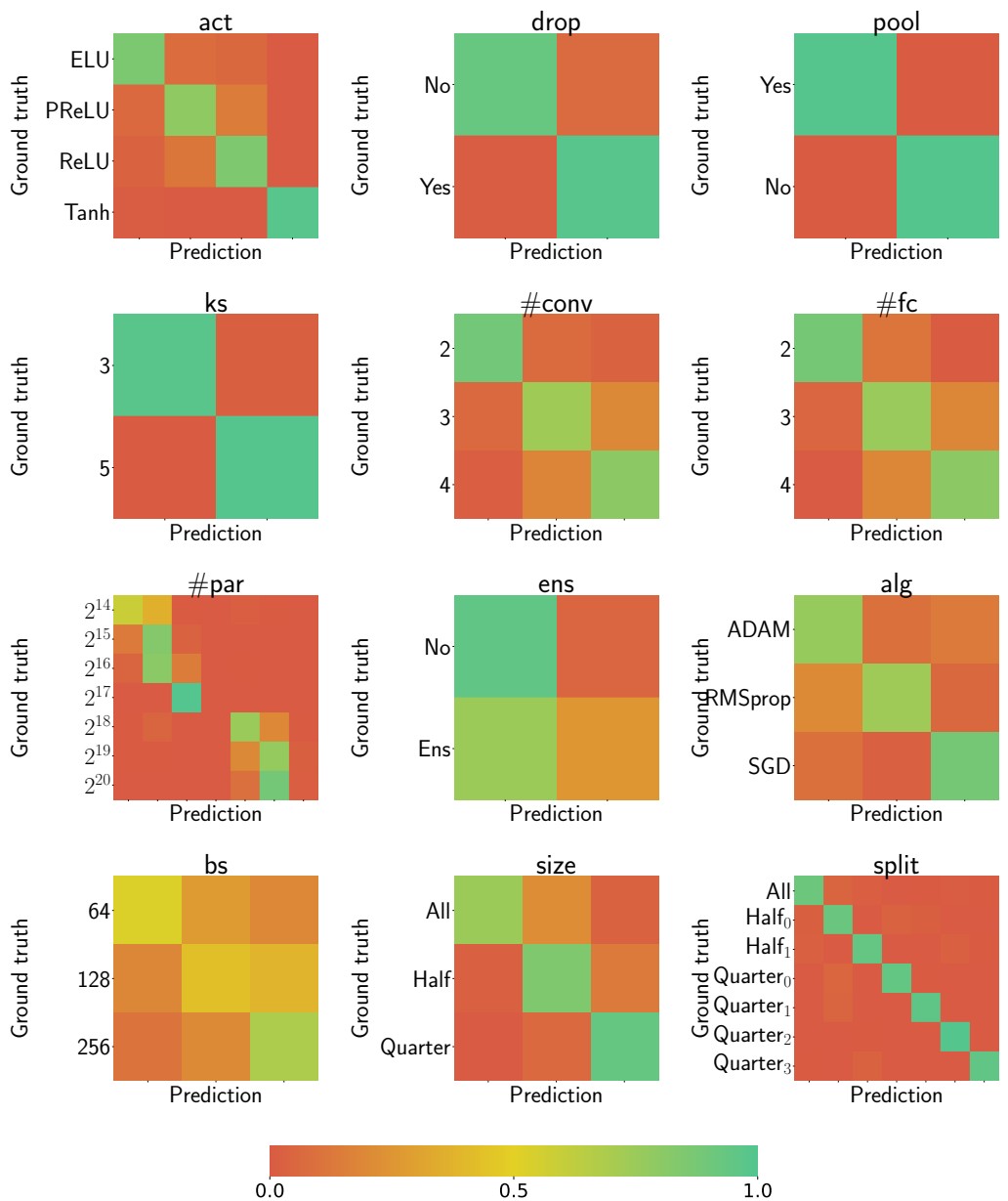

Figure 12: Confusion matrices for `kennen-io`.

