# OpenReview forum: "Towards Reverse-Engineering Black-Box Neural Networks"
_ICLR.cc/2018/Conference — Accept (Poster)_

### Official Review · AnonReviewer2 · 2017-11-24

**Rating:** 7
**Confidence:** 4

**Review:**


-----UPDATE------

Having read the responses from the authors, and the other reviews, I am happy with my rating and maintain that this paper should be accepted.

----------------------



In this paper, the authors trains a large number of MNIST classifier networks with differing attributes (batch-size, activation function, no. layers etc.) and then utilises the inputs and outputs of these networks to predict said attributes successfully. They then show that they are able to use the methods developed to predict the family of Imagenet-trained networks and use this information to improve adversarial attack.

I enjoyed reading this paper. It is a very interesting set up, and a novel idea.

A few comments:

The paper is easy to read, and largely written well. The article is missing from the nouns quite often though so this is something that should be amended. There are a few spelling slip ups ("to a certain extend" --> "to a certain extent", "as will see" --> "as we will see")

It appears that the output for kennen-o is a discrete probability vector for each attribute, where each entry corresponds to a possibility (for example, for "batch-size" it is a length 3 vector where the first entry corresponds to 64, the second 128, and the third 256). What happens if you instead treat it as a regression task, would it then be able to hint at intermediates (a batch size of 96) or extremes (say, 512).

A flaw of this paper is that kennen-i and io appear to require gradients from the network being probed (you do mention this in passing), which realistically you would never have access to. (Please do correct me if I have misunderstood this)

It would be helpful if Section 4 had a paragraph as to your thoughts regarding why certain attributes are easier/harder to predict. Also, the caption for Table 2 could contain more information regarding the network outputs.

You have jumped from predicting 12 attributes on MNIST to 1 attribute on Imagenet. It could be beneficial to do an intermediate experiment (a handful of attributes on a middling task).

I think this paper should be accepted as it is interesting and novel.

Pros
------
- Interesting idea
- Reads well
- Fairly good experimental results

Cons
------
- kennen-i seems like it couldn't be realistically deployed
- lack of an intermediate difficulty task

---

> ### Author Response · Authors · 2018-01-02
> **Response 1/2**
>
> AR{n} = AnonReviewer{n}
>
> We thank all the reviewers for their recognition of the task of whitening black box to be “interesting” and “novel”, and finding the experimental results “interesting” and even “surprising”. In particular, AR2 has commented that “[the task] is a very interesting set up, and a novel idea” and that the paper “should be accepted”. Yet, we find some misunderstandings from the reviewers that could potentially have led to less recognition of the importance & novelty of our work -- we have clarified them here and have updated the paper accordingly. The update is substantial -- 6 more figures & tables, 10s of paragraphs added & updated.
>
> Stressing our contribution again, our results have crucial implications to privacy and security of deep neural network models -- the paper opens an avenue for enhancing the effectiveness of attacks on black boxes. The paper investigates for the first time an important observation that a “relatively broad set of meta parameters” (AR1) can be reliably predicted only from black-box access. We also empirically test our methods in challenging, realistic conditions: What if outputs are single labels? What if there is a big generalisation gap between training models and the test black box? (We will answer AR1’s questions regarding this point.) In addition, kennen-i/io are novel methods that not only learns to interpret the black-box output, but also actively searches for effective query inputs. In particular, contrary to the misunderstanding of AR1 and AR2, kennen-i/io still only requires black-box access to the model at test time. (We will describe in greater detail later.)
>
> We will now answer issues raised by the reviewers one by one.
>
> <<Generalisability of metamodel beyond the training models (AR1)>>
> We also find generalisation problem absolutely important, since in practice DNNs can be trained with diverse “preprocessing, ensembling, [... and] architectural tricks” (AR1). The reviewer has suggested an experiment where the metamodel is trained to predict, say, optimization hyperparameters for a black box model with architecture “D”, when it is only trained on architectures “S,V,B,R” (i.e. generalisation across architecture).
>
> Section 4.2 and table 3 are exactly doing this analysis. In the updated paper, we have added quite some more details of experimental procedure and evaluation details to make them more understandable (experiments themselves are unchanged). We have defined the term “splitting attribute” that denotes the attribute that separates the training and testing models (e.g. architecture family for the example given by AR1). For evaluation, we measure the performance only over non-splitting attributes (e.g. optimization hyperparameters in AR1’s example).
>
> We have shown in the paper that the metamodels do indeed generalise across domain gaps of 1-2 attributes. For example, row 3 of table 3 shows that even when metamodels are trained only on models with #conv<=3 and #fc<=3 (shallower), they can still predict hyperparameters of models with #conv=4 and #fc=4 (deeper) at 80.7% level of the random-split accuracy. The set of experiments in table 3 gives strong evidence that the metamodels do generalise, and this certainly makes the story “more compelling” (AR1). We have also updated table 3 with kennen-o results, following AR3’s suggestion.
>
> AR1 is also wondering about the generalisability of adversarial image perturbation (AIP) attacks across attributes. Our AIP results in section 5.3 are already exhibiting some form of generalisation - we do “leave-one-out cross validation” (section 5.3) evaluation within each family for every AIP evaluation. As the newly added table 7 shows, there exists high intra-family diversity of models in terms of #parameters and #layers. High fooling rates across such a diversity, again, gives evidence that AIPs also do generalise.
>
> <<kennen-i/io requires gradient from the black box (AR2,AR1)>>
> No. All metamodels, including kennen-i/io, *only queries* the test black-box. The requirement for model gradient arises during the training time for kennen-i/io (which is legitimate). We were stressing the fact that kennen-o does not require gradients even from the training models.
>
> Briefly re-describing the procedure of kennen-i/io, at training time they treat the query input as a set of learnable parameters (as for MLP model parameters for kennen-o), and then at test time they feed the *learned* query input to the test black-box model, and read off the outputs.
>
> Kennen-i/io are therefore novel and distinctive methods - they treat *inputs* to a network as a generalisable model parameter. It is indeed quite surprising that the learned input generalises very well to unseen models -- analysing this intriguing phenomenon would be a good future research direction. We have updated the method description in section 3.2 to make our novelty much clearer.

---

> > ### Author Response · Authors · 2018-01-02
> > **Response 2/2**
> >
> > <<“More insights are needed” (AR3)>>
> >  We agree that more clearly determining patterns in the neural network outputs that correlate with the model hyperparameters will shed more light on the interpretability and further applicability of the metamodels -- e.g. “hyperparameter optimization” (AR3). However, we would like to point out that such an analysis deserves a separate paper on its own. The main point of our paper is the *existence* of correlation between model hyperparameters and the outputs and *novel methods* for amplifying and scraping such correlation.
> >
> > To give a glimpse of possible patterns in the model outputs and inner workings of the metamodels, we have updated the paper with two more analyses - one studying patterns in the model outputs (t-SNE visualisation) and the other studying the confusion patterns in the metamodel predictions. See the new section 4.3 for the complete results and discussion. Summarising key observations, t-SNE visualisation of the model outputs show the existence of clusters for some hyperparameters that could explain the good prediction ability of metamodels. In the confusion matrix analysis, we show that the metamodel consistently confuses between similar attributes more often than between dissimilar ones (e.g. confuses more between batch size 64 and 128 than between 64 and 256), for most attribute types. This is a strong indication that the metamodel indeed learns semantically meaningful features rather than mere artifacts.
> >
> > <<On the importance of choice of inputs for kennen-o (AR3)>>
> > For MNIST and ImageNet, the queries were selected (uniform) randomly from the respective validation sets. We have updated the paper with (1) analysis on the importance of the choice of queries for kennen-o and (2) conceptual and empirical comparison between kennen-io and the suggested “optimised choice of queries from validation images” in Appendix section C and figure 6. We provide a brief summary of the experiments and discussion here.
> >
> > The random choice of queries within the validation set turns out to be not critical towards the final performance -- kennen-o trained over 100 independent samples of single queries have the mean average accuracy 42.3%, with standard deviation only 1.2 pp. For more number of queries (10 and 100), the standard deviation of performance among different samples is further reduced (0.7 and 0.5 pp).
> >
> > As AR3 has suggested, one could also search for information-maximising set of queries -- this is an interesting idea left as future work. Instead of solving this potentially difficult combinatorial problem, we have proposed kennen-i/io metamodels that search for the query inputs from the *entire input space*. Empirically, this approach envelops the performance of the 100 kennen-o performances trained with 100 different random query sets at different numbers of queries (figure 6). However, as kennen-i/io submit unnatural input to the model, they could be more detectable. We thus observe a trade-off between performance and detectability of the attack; the proposed idea of maximising query within the natural image domain could potentially relax the trade-off, again an interesting future research direction.
> >
> > <<Lack of intermediate size experiment (AR2)>>
> > We agree that this would be nice to have, but we chose to allocate resources (time and page limit) on (1) extensive analysis on small (therefore efficient) dataset (MNIST) and (2) small number of representative experiments on a more realistic dataset (ImageNet).
> >
> > <<Solving as regression task (AR2)>>
> > Indeed some attributes (e.g. batch size or number of layers) are endowed with natural structures (e.g. batch sizes ‘64’ and ‘128’ are closer than ‘64’ and ‘256’), and sometimes solving the problem as a regression task is more natural. This is an interesting future work. This paper is focused on showing that such attributes can be detected at all, rather than maximising the performance.
> >
> > Finally thank you for the suggestions for improving the paper clarity:
> > Augmenting table 3 with kennen-o (AR3)
> > Explaining ImageNet arch (AR3)
> > Caption for table 2 (AR2)
> > Rationale for the results (AR2)
> > Spelling & grammar (AR2)
> > We have updated the paper according to your suggestions.

---

### Official Review · AnonReviewer1 · 2017-11-27
**interesting approach to adversarial examples but I find generalization thus applicability is hard to assess**

**Rating:** 5
**Confidence:** 4

**Review:**

The paper attempts to study model meta parameter inference e.g. model architecture, optimization, etc using a supervised learning approach. They take three approaches one whereby the target models are evaluated on a fixed set of inputs, one where the access to the gradients is assumed and using that an input is crafted that can be used to infer the target quantities and one where both approaches are combined. The authors also show that these inferred quantities can be used to generate more effective attacks against the targets.

The paper is generally well written and most details for reproducibility are seem enough. I also find the question interesting and the fact that it works on this relatively broad set of meta parameters and under a rigorous train/test split intriguing. It is of course not entirely surprising that the system can be trained but that there is some form of generalization happening.

Aside that I think most system in practical use will be much more different than any a priori enumeration/brute force search for model parameters. I suspect in most cases practical systems will be adapted with many subsequent levels of preprocessing, ensembling, non-standard data and a number of optimization and architectural tricks that are developer dependent. It is really hard to say what a supervised learning meta-model approach such as the one presented in this work have to say about that case.

I have found it hard to understand what table 3 in section 4.2 actually means. It seems to say for instance that a model is trained on 2 and 3 layers then queried with 4 and the accuracy only slightly drops. Accuracy of what ? Is it the other attributes ? Is it somehow that attribute ? if so how can that possibly ?

My main main concern is extrapolation out of the training set which is particularly important here. I don't find enough evidence in 4.2 for that point. One experiment that i would find compelling is to train for instance a meta model on S,V,B,R but not D on imagenet, predict all the attributes except architecture and see how that changes when D is added. If these are better than random and the perturbations are more successful it would be a much more compelling story.

---

> ### Author Response · Authors · 2018-01-02
> **Response 1/2**
>
> AR{n} = AnonReviewer{n}
>
> We thank all the reviewers for their recognition of the task of whitening black box to be “interesting” and “novel”, and finding the experimental results “interesting” and even “surprising”. In particular, AR2 has commented that “[the task] is a very interesting set up, and a novel idea” and that the paper “should be accepted”. Yet, we find some misunderstandings from the reviewers that could potentially have led to less recognition of the importance & novelty of our work -- we have clarified them here and have updated the paper accordingly. The update is substantial -- 6 more figures & tables, 10s of paragraphs added & updated.
>
> Stressing our contribution again, our results have crucial implications to privacy and security of deep neural network models -- the paper opens an avenue for enhancing the effectiveness of attacks on black boxes. The paper investigates for the first time an important observation that a “relatively broad set of meta parameters” (AR1) can be reliably predicted only from black-box access. We also empirically test our methods in challenging, realistic conditions: What if outputs are single labels? What if there is a big generalisation gap between training models and the test black box? (We will answer AR1’s questions regarding this point.) In addition, kennen-i/io are novel methods that not only learns to interpret the black-box output, but also actively searches for effective query inputs. In particular, contrary to the misunderstanding of AR1 and AR2, kennen-i/io still only requires black-box access to the model at test time. (We will describe in greater detail later.)
>
> We will now answer issues raised by the reviewers one by one.
>
> <<Generalisability of metamodel beyond the training models (AR1)>>
> We also find generalisation problem absolutely important, since in practice DNNs can be trained with diverse “preprocessing, ensembling, [... and] architectural tricks” (AR1). The reviewer has suggested an experiment where the metamodel is trained to predict, say, optimization hyperparameters for a black box model with architecture “D”, when it is only trained on architectures “S,V,B,R” (i.e. generalisation across architecture).
>
> Section 4.2 and table 3 are exactly doing this analysis. In the updated paper, we have added quite some more details of experimental procedure and evaluation details to make them more understandable (experiments themselves are unchanged). We have defined the term “splitting attribute” that denotes the attribute that separates the training and testing models (e.g. architecture family for the example given by AR1). For evaluation, we measure the performance only over non-splitting attributes (e.g. optimization hyperparameters in AR1’s example).
>
> We have shown in the paper that the metamodels do indeed generalise across domain gaps of 1-2 attributes. For example, row 3 of table 3 shows that even when metamodels are trained only on models with #conv<=3 and #fc<=3 (shallower), they can still predict hyperparameters of models with #conv=4 and #fc=4 (deeper) at 80.7% level of the random-split accuracy. The set of experiments in table 3 gives strong evidence that the metamodels do generalise, and this certainly makes the story “more compelling” (AR1). We have also updated table 3 with kennen-o results, following AR3’s suggestion.
>
> AR1 is also wondering about the generalisability of adversarial image perturbation (AIP) attacks across attributes. Our AIP results in section 5.3 are already exhibiting some form of generalisation - we do “leave-one-out cross validation” (section 5.3) evaluation within each family for every AIP evaluation. As the newly added table 7 shows, there exists high intra-family diversity of models in terms of #parameters and #layers. High fooling rates across such a diversity, again, gives evidence that AIPs also do generalise.
>
> <<kennen-i/io requires gradient from the black box (AR2,AR1)>>
> No. All metamodels, including kennen-i/io, *only queries* the test black-box. The requirement for model gradient arises during the training time for kennen-i/io (which is legitimate). We were stressing the fact that kennen-o does not require gradients even from the training models.
>
> Briefly re-describing the procedure of kennen-i/io, at training time they treat the query input as a set of learnable parameters (as for MLP model parameters for kennen-o), and then at test time they feed the *learned* query input to the test black-box model, and read off the outputs.
>
> Kennen-i/io are therefore novel and distinctive methods - they treat *inputs* to a network as a generalisable model parameter. It is indeed quite surprising that the learned input generalises very well to unseen models -- analysing this intriguing phenomenon would be a good future research direction. We have updated the method description in section 3.2 to make our novelty much clearer.

---

> > ### Author Response · Authors · 2018-01-02
> > **Response 2/2**
> >
> > <<“More insights are needed” (AR3)>>
> >  We agree that more clearly determining patterns in the neural network outputs that correlate with the model hyperparameters will shed more light on the interpretability and further applicability of the metamodels -- e.g. “hyperparameter optimization” (AR3). However, we would like to point out that such an analysis deserves a separate paper on its own. The main point of our paper is the *existence* of correlation between model hyperparameters and the outputs and *novel methods* for amplifying and scraping such correlation.
> >
> > To give a glimpse of possible patterns in the model outputs and inner workings of the metamodels, we have updated the paper with two more analyses - one studying patterns in the model outputs (t-SNE visualisation) and the other studying the confusion patterns in the metamodel predictions. See the new section 4.3 for the complete results and discussion. Summarising key observations, t-SNE visualisation of the model outputs show the existence of clusters for some hyperparameters that could explain the good prediction ability of metamodels. In the confusion matrix analysis, we show that the metamodel consistently confuses between similar attributes more often than between dissimilar ones (e.g. confuses more between batch size 64 and 128 than between 64 and 256), for most attribute types. This is a strong indication that the metamodel indeed learns semantically meaningful features rather than mere artifacts.
> >
> > <<On the importance of choice of inputs for kennen-o (AR3)>>
> > For MNIST and ImageNet, the queries were selected (uniform) randomly from the respective validation sets. We have updated the paper with (1) analysis on the importance of the choice of queries for kennen-o and (2) conceptual and empirical comparison between kennen-io and the suggested “optimised choice of queries from validation images” in Appendix section C and figure 6. We provide a brief summary of the experiments and discussion here.
> >
> > The random choice of queries within the validation set turns out to be not critical towards the final performance -- kennen-o trained over 100 independent samples of single queries have the mean average accuracy 42.3%, with standard deviation only 1.2 pp. For more number of queries (10 and 100), the standard deviation of performance among different samples is further reduced (0.7 and 0.5 pp).
> >
> > As AR3 has suggested, one could also search for information-maximising set of queries -- this is an interesting idea left as future work. Instead of solving this potentially difficult combinatorial problem, we have proposed kennen-i/io metamodels that search for the query inputs from the *entire input space*. Empirically, this approach envelops the performance of the 100 kennen-o performances trained with 100 different random query sets at different numbers of queries (figure 6). However, as kennen-i/io submit unnatural input to the model, they could be more detectable. We thus observe a trade-off between performance and detectability of the attack; the proposed idea of maximising query within the natural image domain could potentially relax the trade-off, again an interesting future research direction.
> >
> > <<Lack of intermediate size experiment (AR2)>>
> > We agree that this would be nice to have, but we chose to allocate resources (time and page limit) on (1) extensive analysis on small (therefore efficient) dataset (MNIST) and (2) small number of representative experiments on a more realistic dataset (ImageNet).
> >
> > <<Solving as regression task (AR2)>>
> > Indeed some attributes (e.g. batch size or number of layers) are endowed with natural structures (e.g. batch sizes ‘64’ and ‘128’ are closer than ‘64’ and ‘256’), and sometimes solving the problem as a regression task is more natural. This is an interesting future work. This paper is focused on showing that such attributes can be detected at all, rather than maximising the performance.
> >
> > Finally thank you for the suggestions for improving the paper clarity:
> > Augmenting table 3 with kennen-o (AR3)
> > Explaining ImageNet arch (AR3)
> > Caption for table 2 (AR2)
> > Rationale for the results (AR2)
> > Spelling & grammar (AR2)
> > We have updated the paper according to your suggestions.

---

### Official Review · AnonReviewer3 · 2017-11-28
**interesting results, but more insights would be helpful**

**Rating:** 7
**Confidence:** 3

**Review:**

The basic idea is to train a neural network to predict various hyperparameters of a classifier from input-output pairs for that classifier (kennen-o approach). It is surprising that some of these hyperparameters can even be predicted with more than chance accuracy. As a simple example, it's possible that there are values of batch size for which the classifiers may become indistinguishable, yet Table 2 shows that batch size can be predicted with much higher accuracy than chance. It would be good to provide insights into under what conditions and why hyperparameters can be predicted accurately. That would make the results much more interesting, and may even turn out to be useful for other problems, such as hyperparameter optimization.

The selection of the queries for kennen-o is not explained. What is the procedure for selecting the queries? How sensitive is the performance of kennen-o to the choice of the queries? One would expect that there is significant sensitivity, in which case it may even make sense to consider learning to select a sequence of queries to maximize accuracy.

In table 3, it would be useful to show the results for kennen-o as well, because Split-E seems to be the more realistic problem setting and kennen-o seems to be a more realistic attack than kennen-i or kennen-io.

In the ImageNet classifier family prediction, how different are the various families from each other? Without going through all the references, it is difficult to get a sense of the difficulty of the prediction task for a non-computer-vision reader.

Overall the results seem interesting, but without more insights it's difficult to judge how generally useful they are.

---

> ### Author Response · Authors · 2018-01-02
> **Response 1/2**
>
> AR{n} = AnonReviewer{n}
>
> We thank all the reviewers for their recognition of the task of whitening black box to be “interesting” and “novel”, and finding the experimental results “interesting” and even “surprising”. In particular, AR2 has commented that “[the task] is a very interesting set up, and a novel idea” and that the paper “should be accepted”. Yet, we find some misunderstandings from the reviewers that could potentially have led to less recognition of the importance & novelty of our work -- we have clarified them here and have updated the paper accordingly. The update is substantial -- 6 more figures & tables, 10s of paragraphs added & updated.
>
> Stressing our contribution again, our results have crucial implications to privacy and security of deep neural network models -- the paper opens an avenue for enhancing the effectiveness of attacks on black boxes. The paper investigates for the first time an important observation that a “relatively broad set of meta parameters” (AR1) can be reliably predicted only from black-box access. We also empirically test our methods in challenging, realistic conditions: What if outputs are single labels? What if there is a big generalisation gap between training models and the test black box? (We will answer AR1’s questions regarding this point.) In addition, kennen-i/io are novel methods that not only learns to interpret the black-box output, but also actively searches for effective query inputs. In particular, contrary to the misunderstanding of AR1 and AR2, kennen-i/io still only requires black-box access to the model at test time. (We will describe in greater detail later.)
>
> We will now answer issues raised by the reviewers one by one.
>
> <<Generalisability of metamodel beyond the training models (AR1)>>
> We also find generalisation problem absolutely important, since in practice DNNs can be trained with diverse “preprocessing, ensembling, [... and] architectural tricks” (AR1). The reviewer has suggested an experiment where the metamodel is trained to predict, say, optimization hyperparameters for a black box model with architecture “D”, when it is only trained on architectures “S,V,B,R” (i.e. generalisation across architecture).
>
> Section 4.2 and table 3 are exactly doing this analysis. In the updated paper, we have added quite some more details of experimental procedure and evaluation details to make them more understandable (experiments themselves are unchanged). We have defined the term “splitting attribute” that denotes the attribute that separates the training and testing models (e.g. architecture family for the example given by AR1). For evaluation, we measure the performance only over non-splitting attributes (e.g. optimization hyperparameters in AR1’s example).
>
> We have shown in the paper that the metamodels do indeed generalise across domain gaps of 1-2 attributes. For example, row 3 of table 3 shows that even when metamodels are trained only on models with #conv<=3 and #fc<=3 (shallower), they can still predict hyperparameters of models with #conv=4 and #fc=4 (deeper) at 80.7% level of the random-split accuracy. The set of experiments in table 3 gives strong evidence that the metamodels do generalise, and this certainly makes the story “more compelling” (AR1). We have also updated table 3 with kennen-o results, following AR3’s suggestion.
>
> AR1 is also wondering about the generalisability of adversarial image perturbation (AIP) attacks across attributes. Our AIP results in section 5.3 are already exhibiting some form of generalisation - we do “leave-one-out cross validation” (section 5.3) evaluation within each family for every AIP evaluation. As the newly added table 7 shows, there exists high intra-family diversity of models in terms of #parameters and #layers. High fooling rates across such a diversity, again, gives evidence that AIPs also do generalise.
>
> <<kennen-i/io requires gradient from the black box (AR2,AR1)>>
> No. All metamodels, including kennen-i/io, *only queries* the test black-box. The requirement for model gradient arises during the training time for kennen-i/io (which is legitimate). We were stressing the fact that kennen-o does not require gradients even from the training models.
>
> Briefly re-describing the procedure of kennen-i/io, at training time they treat the query input as a set of learnable parameters (as for MLP model parameters for kennen-o), and then at test time they feed the *learned* query input to the test black-box model, and read off the outputs.
>
> Kennen-i/io are therefore novel and distinctive methods - they treat *inputs* to a network as a generalisable model parameter. It is indeed quite surprising that the learned input generalises very well to unseen models -- analysing this intriguing phenomenon would be a good future research direction. We have updated the method description in section 3.2 to make our novelty much clearer.

---

> > ### Author Response · Authors · 2018-01-02
> > **Response 2/2**
> >
> > <<“More insights are needed” (AR3)>>
> >  We agree that more clearly determining patterns in the neural network outputs that correlate with the model hyperparameters will shed more light on the interpretability and further applicability of the metamodels -- e.g. “hyperparameter optimization” (AR3). However, we would like to point out that such an analysis deserves a separate paper on its own. The main point of our paper is the *existence* of correlation between model hyperparameters and the outputs and *novel methods* for amplifying and scraping such correlation.
> >
> > To give a glimpse of possible patterns in the model outputs and inner workings of the metamodels, we have updated the paper with two more analyses - one studying patterns in the model outputs (t-SNE visualisation) and the other studying the confusion patterns in the metamodel predictions. See the new section 4.3 for the complete results and discussion. Summarising key observations, t-SNE visualisation of the model outputs show the existence of clusters for some hyperparameters that could explain the good prediction ability of metamodels. In the confusion matrix analysis, we show that the metamodel consistently confuses between similar attributes more often than between dissimilar ones (e.g. confuses more between batch size 64 and 128 than between 64 and 256), for most attribute types. This is a strong indication that the metamodel indeed learns semantically meaningful features rather than mere artifacts.
> >
> > <<On the importance of choice of inputs for kennen-o (AR3)>>
> > For MNIST and ImageNet, the queries were selected (uniform) randomly from the respective validation sets. We have updated the paper with (1) analysis on the importance of the choice of queries for kennen-o and (2) conceptual and empirical comparison between kennen-io and the suggested “optimised choice of queries from validation images” in Appendix section C and figure 6. We provide a brief summary of the experiments and discussion here.
> >
> > The random choice of queries within the validation set turns out to be not critical towards the final performance -- kennen-o trained over 100 independent samples of single queries have the mean average accuracy 42.3%, with standard deviation only 1.2 pp. For more number of queries (10 and 100), the standard deviation of performance among different samples is further reduced (0.7 and 0.5 pp).
> >
> > As AR3 has suggested, one could also search for information-maximising set of queries -- this is an interesting idea left as future work. Instead of solving this potentially difficult combinatorial problem, we have proposed kennen-i/io metamodels that search for the query inputs from the *entire input space*. Empirically, this approach envelops the performance of the 100 kennen-o performances trained with 100 different random query sets at different numbers of queries (figure 6). However, as kennen-i/io submit unnatural input to the model, they could be more detectable. We thus observe a trade-off between performance and detectability of the attack; the proposed idea of maximising query within the natural image domain could potentially relax the trade-off, again an interesting future research direction.
> >
> > <<Lack of intermediate size experiment (AR2)>>
> > We agree that this would be nice to have, but we chose to allocate resources (time and page limit) on (1) extensive analysis on small (therefore efficient) dataset (MNIST) and (2) small number of representative experiments on a more realistic dataset (ImageNet).
> >
> > <<Solving as regression task (AR2)>>
> > Indeed some attributes (e.g. batch size or number of layers) are endowed with natural structures (e.g. batch sizes ‘64’ and ‘128’ are closer than ‘64’ and ‘256’), and sometimes solving the problem as a regression task is more natural. This is an interesting future work. This paper is focused on showing that such attributes can be detected at all, rather than maximising the performance.
> >
> > Finally thank you for the suggestions for improving the paper clarity:
> > Augmenting table 3 with kennen-o (AR3)
> > Explaining ImageNet arch (AR3)
> > Caption for table 2 (AR2)
> > Rationale for the results (AR2)
> > Spelling & grammar (AR2)
> > We have updated the paper according to your suggestions.

---

> > > ### Comment · AnonReviewer3 · 2018-01-25
> > > **section 4.3 is not convincing**
> > >
> > > Thanks to the authors for the extensive response with further results and analysis. Table 7 was very helpful in understanding how similar/different various architectures are, and the expanded table 3 was helpful in evaluating kennen-o in the extrapolation setting. I didn't find the results in section 4.3 to be convincing or to provide further insight into why it's possible at all to predict hyperparameters with better than chance accuracy, except for perhaps the act hyperparameter.
> > >
> > > I still find it mysterious that, even though the relationship between the hyperparameters and the query input-output pairs is obviously highly nonlinear (an iterative training process to produce a nonlinear function, followed by applying that nonlinear function to the query input to produce the corresponding output), that relationship can be inverted to the accuracy level shown in the paper. Nevertheless I think this work can benefit from broader attention from the community, which may motivate others to try out the proposed approaches on more datasets to get more insights. So I'm changing my rating to 7 despite my skepticism.

---

### Author Response · Authors · 2018-01-04
**Paper revision**

The paper has been updated during the rebuttal period, following suggestions from the reviewers. We only briefly list the updates here; for more detailed information, please see the "Response k/2" (k=1 or 2) comments to each reviewer.

1. Section 3.2 (metamodel method description)
Updated description for kennen-i and kennen-io -- they do not require gradients from black box. (AR1,2)

2. Table 2 (summary table for metamodel results on MNIST classifiers)
Definition of "prob", "ranking", and "1 label" added. (AR2)

3. Section 4 (metamodel results on MNIST classifiers)
More detailed rationales for the results. (AR2)

4. Section 4.2 (extrapolation experiments for metamodels)
Detailed description of experimental setup and evaluation metric. (AR1)

5. Table 3 (extrapolation experiments for metamodels)
kennen-o results added. (AR3)

6. Section 4.3 & D, figures 9-12 (more insights on why and how metamodel works)
Newly added section & figures containing t-SNE visualisation of neural net outputs and confusion matrix analysis for metamodel outputs. (AR3)

7. Section C, figure 6 (finding optimal set of queries for kennen-o)
New section containing experiments and discussion on finding the optimal set of queries for kennen-o. (AR3)

8. Table 7 (architecture profiles for 19 ImageNet classifiers used in Section 5)
New table showing intra-family diversity and inter-family similarity of considered ImageNet classifiers. (AR3)

9. Entire paper - spelling & grammar errors. (AR2)

---

### Decision · Program_Chairs · 2018-01-29
**ICLR 2018 Conference Acceptance Decision**

**Decision:**

Accept (Poster)

**Comment:**

Novel way of analyzing neural networks to predict NN attributes such as architecture, training method, batch size etc. And the method works surprisingly good on the MNIST and ImageNet.